# The application of artificial intelligence to support biliary atresia screening by ultrasound images: A study based on deep learning models

**Fang-Rong Hsu**[1], **Sheng-Tong Dai**[1,2], **Chia-Man Chou**[3,4,5], **Sheng-Yang Huang**[3,4,5]*

**1** Department of Information Engineering and Computer Science, Feng Chia University, Taichung City, Taiwan, **2** Institute of Biomedical Engineering and Nanomedicine, National Health Research Institutes, Miaoli County, Taiwan, **3** Division of Pediatric Surgery, Department of Surgery, Taichung Veterans General Hospital, Taichung City, Taiwan, **4** School of Medicine, College of Medicine, National Yang Ming Chiao Tung University, Taipei City, Taiwan, **5** Department of Post-Baccalaureate Medicine, College of Medicine, National Chung Hsing University, Taichung City, Taiwan

* drugholic@vghtc.gov.tw

**Data Availability Statement:** All relevant data are within the manuscript and its Supporting Information files.

## Abstract

### Purpose

Early confirmation or ruling out biliary atresia (BA) is essential for infants with delayed onset of jaundice. In the current practice, percutaneous liver biopsy and intraoperative cholangiography (IOC) remain the golden standards for diagnosis. In Taiwan, the diagnostic methods are invasive and can only be performed in selective medical centers. However, referrals from primary physicians and local pediatricians are often delayed because of lacking clinical suspicions. Ultrasounds (US) are common screening tools in local hospitals and clinics, but the pediatric hepatobiliary US particularly requires well-trained imaging personnel. The meaningful comprehension of US is highly dependent on individual experience. For screening BA through human observation on US images, the reported sensitivity and specificity were achieved by pediatric radiologists, pediatric hepatobiliary experts, or pediatric surgeons. Therefore, this research developed a tool based on deep learning models for screening BA to assist pediatric US image reading by general physicians and pediatricians.

### Methods

De-identified hepatobiliary US images of 180 patients from Taichung Veterans General Hospital were retrospectively collected under the approval of the Institutional Review Board. Herein, the top network models of ImageNet Large Scale Visual Recognition Competition and other network models commonly used for US image recognition were included for further study to classify US images as BA or non-BA. The performance of different network models was expressed by the confusion matrix and receiver operating characteristic curve. There were two methods proposed to solve disagreement by US image classification of a single patient. The first and second methods were the positive-dominance law and threshold

**Funding:** This study was supported by collaboration research project TCVGH-FCU1108202 from Taichung Veterans General Hospital (https://www.vghtc.gov.tw/) and Feng Chia University (https://www.fcu.edu.tw/) to F. R. Hsu and C. M. Chou. The funders had no role in study design, data collection and analysis, decision to publish, or preparation of the manuscript.

**Competing interests:** The authors have declared that no competing interests exist.

**Abbreviations:** BA, Biliary atresia; IOC, Intraoperative cholangiography; PPV, Positive predictive value; US, Ultrasound; CNN, Convolutional neural network; RNN, Recurrent neural network; ReLU, Rectified linear unit; SE, Squeeze-and-Excitation; ROI, Region of interest; PIQE, Perception-based Image Quality Evaluator.

law. During the study, the US images of three successive patients suspected to have BA were classified by the trained models.

## Results

Among all included patients contributing US images, 41 patients were diagnosed with BA by surgical intervention and 139 patients were either healthy controls or had non-BA diagnoses. In this study, a total of 1,976 original US images were enrolled. Among them, 417 and 1,559 raw images were from patients with BA and without BA, respectively. Meanwhile, ShuffleNet achieved the highest accuracy of 90.56% using the same training parameters as compared with other network models. The sensitivity and specificity were 67.83% and 96.76%, respectively. In addition, the undesired false-negative prediction was prevented by applying positive-dominance law to interpret different images of a single patient with an acceptable false-positive rate, which was 13.64%. For the three consecutive patients with delayed obstructive jaundice with IOC confirmed diagnoses, ShuffleNet achieved accurate diagnoses in two patients.

## Conclusion

The current study provides a screening tool for identifying possible BA by hepatobiliary US images. The method was not designed to replace liver biopsy or IOC, but to decrease human error for interpretations of US. By applying the positive-dominance law to ShuffleNet, the false-negative rate and the specificities were 0 and 86.36%, respectively. The trained deep learning models could aid physicians other than pediatric surgeons, pediatric gastroenterologists, or pediatric radiologists, to prevent misreading pediatric hepatobiliary US images. The current artificial intelligence (AI) tool is helpful for screening BA in the real world.

## Introduction

Prolonged jaundice in neonates and infants is often encountered by pediatricians and pediatric surgeons. The common types include hemolytic jaundice, infectious jaundice, breast milk jaundice, obstructive jaundice, and rare autoimmune diseases [1, 2]. If the serum direct bilirubin level exceeds 1.0 mg/dL or the ratio of direct to indirect bilirubin level exceeds 15%–20% [3], obstructive jaundice is impressed and early surgical intervention is often required. Biliary atresia (BA) is the most common etiology of obstructive jaundice for neonates and infants. In general, surgery before 60 days of age is considered beneficial. Thus, early suspicion, sensitive screening tools, and accurate diagnostic methods for BA are highly demanded. Basic approaches for obstructive jaundice include infant stool card screening, capillary heel stick sampling, and transcutaneous bilirubinometer. The stool card screening is a simple tool for parents to detect abnormal stool color. In addition, the high sensitivity and low PPV of stool cards represent appropriate screening values [4]. Meanwhile, the capillary heel stick sampling and transcutaneous bilirubinometer are basic tools to evaluate serum bilirubin levels instead of direct venous blood sampling. The two less invasive tools are developed to replace invasive blood sampling for infants but not for screening BA [5].

However, liver biopsy and intraoperative cholangiography (IOC) remain the gold standard for BA diagnosis. The two invasive procedures require judicious clinical decisions and can

only be performed in exclusive centers. Sensitivity (50%–100%) and specificity (66.7%–100%) of percutaneous liver biopsy are not ideal because of technical issues such as sampling error and inaccurate pathological interpretation [6]. IOC can only be performed by pediatric surgeons who can perform simultaneous Kasai portoenterostomy, thus the procedure should be preserved for patients with the most possibility of BA. Meanwhile, for patients without BA, the invasive diagnostic tools are often considered unnecessary by parents and could lead to additional complications. Some advanced non-invasive image studies have been used for early screening and diagnosis. Hepatobiliary scintigraphy has a sensitivity of 98.1%–99.2%, however, the specificity is as low as 68.5%–72.2% [7]. Moreover, the isotopes for the examination are no longer available in most regions and countries. Magnetic resonance imaging (MRI) presents good sensitivity and specificity of up to 98% [8]. However, the procedure requires sedation, and MRI equipment with better accuracy of infant images is not common in local hospitals. Hepatobiliary ultrasound (US) has the advantages of feasibility and non-invasiveness. Screening of BA based on the US depends on triangular cord sign and gallbladder anomaly in fasting status [2]. Furthermore, a triangular cord sign defines fibrosis of the extrahepatic bile duct over the hepatic hilum, and a hyperechogenic area posterior to the hilum vessels could be visible. The sensitivity and specificity of the triangular cord sign are 61%–84% and 95%–99%, respectively. Typically, the gallbladder size should be larger during fasting and smaller after feeding. Patients with BA often had invisible gallbladder and no size change after feeding. The sensitivity and specificity of US gallbladder size for BA are 76%–91% and 81%–97%, respectively. Collectively, the US has a sensitivity and a specificity of 70%–99% and 79%–94% [9]. However, the high diagnostic performance of the US can only be achieved by well-trained pediatric surgeons, pediatricians, and pediatric radiologists. Image quality is highly user-dependent, and the inter-observer difference is expected, particularly for primary physicians and pediatricians other than experts.

This study took the work of Kuo et al. [10] as a reference and successfully established a US-based deep learning model for BA screening without human interpretation. Hepatobiliary US images of patients with and without BA were retrospectively collected as a raw database. There were no marking or prior identification of US details, such as triangular cord sign, gallbladder size, hepatic hilum structure, or intrahepatic bile ducts made on the raw images to exclude expertise requirements, prevent human error, and expand further applications without aid from specialists. The new artificial intelligence (AI) model improves the screening process for BA and earlier detects disease. For patients with obstructive jaundice rather than BA, this model may prevent unnecessary surgeries or liver biopsies at outpatient service. At present, scanty reports discussed BA diagnosis by AI-based US image classification [11]. In this study, a reliable model was established to facilitate the non-invasive early screening of BA.

## Methods

### Study structure

This research mainly included four parts, namely data collection and labeling, ultrasonic image data preprocessing, CNN network model training and classification, evaluation, and optimization network. The workflow chart is shown in Fig 1. The details of each step will be addressed.

### Data collection and labeling

Patients with accessible pre-operative hepatobiliary US images who underwent Kasai portoenterostomy for BA in Taichung Veterans General Hospital before December 31, 2020, were retrospectively collected. For comparison, continuous non-BA patients with hepatobiliary US

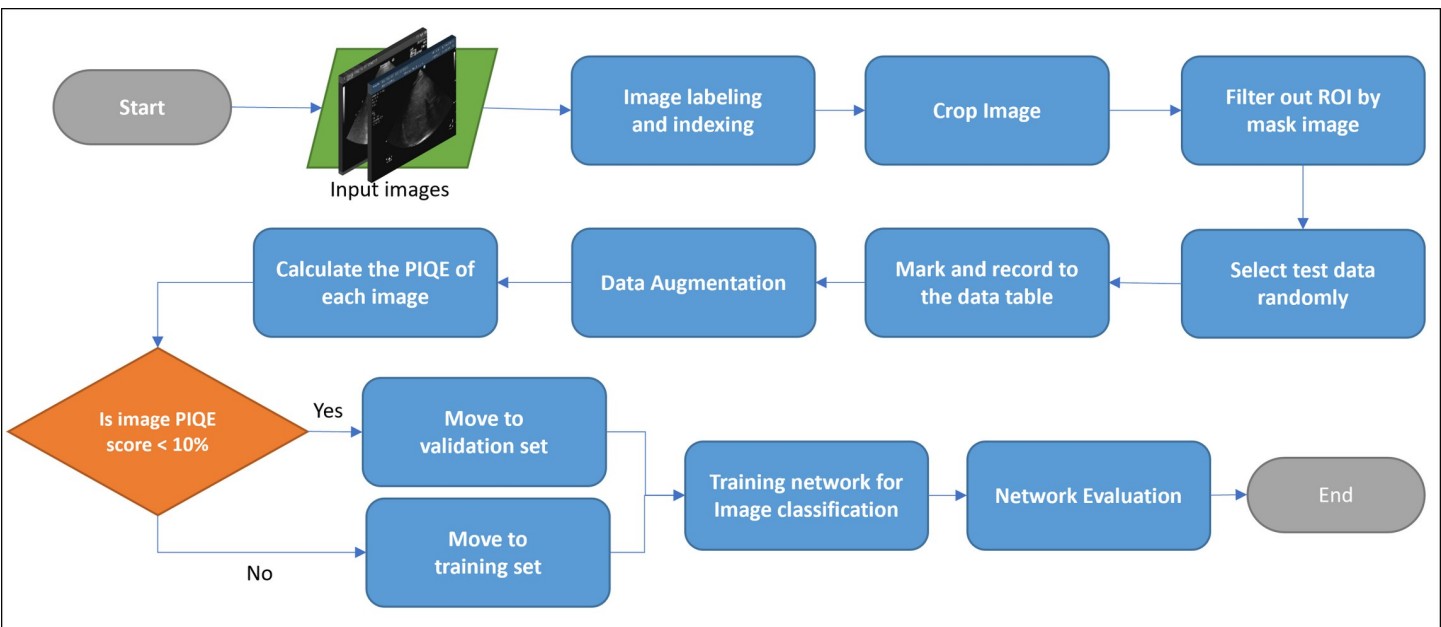

**Fig 1. Workflow chart of neural network training for US image recognition.** ROI: range of interest. PIQE: perception-based quality evaluator.

images obtained between January 1, 2020–September 1, 2020, were also gathered. Meanwhile, patients older than one year, without accessible US images, or who had hepatobiliary tract surgery before the US were excluded. Patients' characteristics, the number of patients, and US images are listed in Table 1. Patients with BA were younger than patients without BA, however, the gender distribution and body weight were the same in the two groups. Serum alanine aminotransferase (ALT) and γ-glutamyltransferase (GGT) were high in BA patients. The study

**Table 1. Patients' characteristics, patient number, and US images amounts of BA and non-BA groups.**

|  | Biliary atresia | Non-biliary atresia |  |
| --- | --- | --- | --- |
| **Number of** | | | **Total** |
| **Patients** | 41 | 139 | 180 |
| **Original US images** | 417 | 1,559 | 1,976 |
| **Included images** | 350 | 1,416 | 1,766 |
| **Characteristics** | | | **p-value** |
| **Female** | 53.7% | 43.9% | 0.271 |
| **Age (month)** | 1.8 (1.5–2.2) | 2.9 (2.5–3.4) | 0.035 |
| **Body weight (kg)** | 4.37 (3.99–4.76) | 4.84 (4.42–5.26) | 0.298 |
| **Total bilirubin (mg/dL)** | 8.87 (7.61–10.12) | 4.95 (3.81–6.10) | <0.001 |
| **Direct bilirubin (mg/dL)** | 4.19 (3.70–4.68) | 1.06 (0.70–1.42) | < 0.001 |
| **Aspartate aminotransferase, AST (U/L)** | 156.3 (119.0–193.7) | 94.1 (42.0–146.3) | 0.125 |
| **Alanine aminotransferase, ALT (U/L)** | 117.7 (84.7–150.7) | 39.0 (27.9–50.0) | < 0.001 |
| **Alkaline phosphatase, ALP (U/L)** | 717.0 (596.3–837.6) | 544.6 (359.2–730.0) | 0.098 |
| **γ-glutamyltransferase, GGT (U/L)** | 768.4 (578.4–958.4) | 138.5 (83.6–193.3) | < 0.001 |

Images with poor quality and Doppler mode were excluded before entering training or the test set. Data presentation: mean (95% confidence interval) for continuous variables. Significance tests: Chi-squared test for categorical variables and t-test for continuous variables.

was approved by the Institutional Review Board (IRB) II of Taichung Veterans General Hospital (consent number CE21036B3). Images for the study were reviewed by two authors (S.Y.H. and C.M.C.) independently, and any disagreement was resolved by discussion. Images from different angles observing the hepatobiliary region were ultimately collected; thus, one patient would have various US images for analysis.

This study is conducted to observe images of hepatobiliary US to determine whether or not the patient has BA. The brightness mode (B-mode) prevents overfitting and facilitates future application (Fig 2A). Color Doppler images would be excluded in further training (Fig 2B). Four angles of observation of hepatobiliary US images were routinely obtained. The first involved the right liver lobe, right kidney, and right psoas muscle. Meanwhile, the second included the gallbladder and portal vein. The third observed the hepatic vein, inferior vena cava, and portal vein, and the final view was focused on the portal vein, common bile duct, and gallbladder. All US examinations were performed by one experienced sonographer using a Philips HD 11 XE ultrasound scanner and an S12-4 sector-array transducer (Philips Medical, Eindhoven, The Netherlands).

## Preprocessing of ultrasound images

For clinical references, some texts were marked in the original US images, such as "NPO", "Feeding", and "GB" for *nil per os* (fasting), post-feeding, and gallbladder, respectively. In the preprocessing stage, the texts were filtered out to acquire original US images and avoid texts appearing in the region of interest (ROI). Then, after obtaining the original hepatobiliary US images selected by two pediatric surgeons, the first step was to crop the target range. Herein, a custom mask was used to filter out unwanted ROI information, preserve the fan-shaped area in the US images, delete the marginal black area, and neglect information on equipment descriptions. There was no further specific identification of anatomical landmarks or signs for BA diagnosis made to simulate routine hepatobiliary US evaluated by non-experts. Subsequently, images were written into the data table of the corresponding category while the index, file name, and location of images were recorded. The quality of the image was important for the performance of the neural network model, and the inconsistency of the image quality during the training and verification phases would decrease the performance of the network model

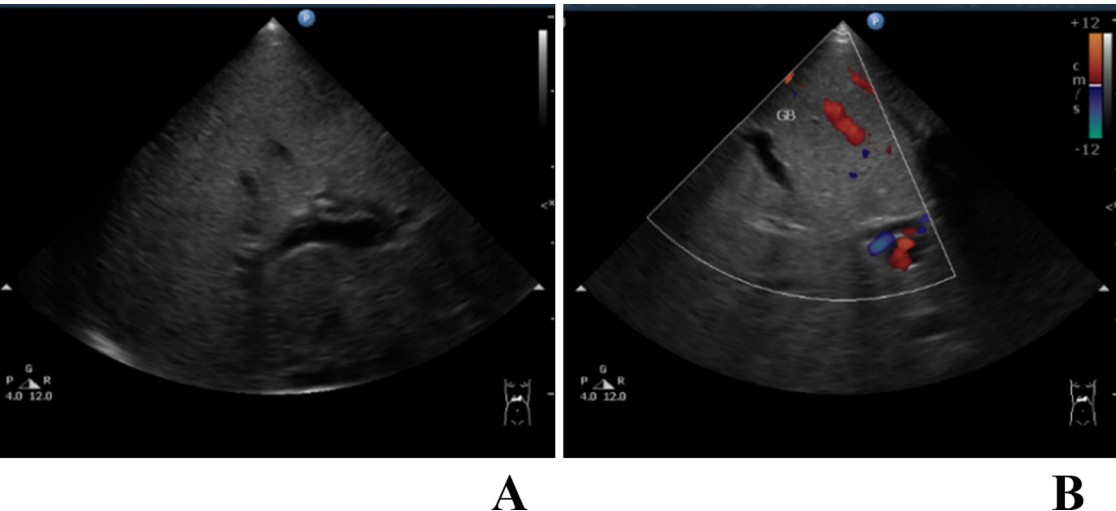

**Fig 2. Examples of US images from a patient of BA.** A. an US image in brightness mode. B. an US image with annotation in Dopplermode. GB: gallbladder.

[12]. Therefore, a Perception-based Image Quality Evaluator (PIQE) was used to screen the process [10, 13]. In addition, quality metrics provided objective image quality scores, and two types of algorithms could be used, full-reference and no-reference, respectively. The former compared the input images with the original reference images without distortion whereas the latter compared the statistical features of the input images with a set of features derived from the image database. This study used PIQE by block-wise distortion estimation to calculate the non-reference image quality score [13].

The PIQE score calculated the mean subtracted contrast normalized (MSCN) coefficient of each pixel in the image. The first step was to calculate the parameters of local mean removal and divisive normalization for the input image. This step extracted the natural scene statistics (NSS) feature. Afterward, only the spatially active blocks were evaluated for the quality score. Image distortion could be divided into three categories: blockiness, blur, and noise. The spatially active block distortion could be divided into two types of processing: noticeable distortion and white noise. Then, the variance of the block is used to measure the two kinds of distortions, and the quality of the whole image was expressed in a percentage ≤ 100%. In this study, the cutoff value of PIQE scores to determine good quality is ≤ 10%.

As the test set, 11 patients with BA (143 US images) and 45 patients without BA (525 US images) were randomly selected with a total of 668 US images. The randomization process was performed by the blinded author (S.T.D.) who had no access to patients' characteristics or any clinical information. In addition, some images were randomly selected as the test set among the included US images. The rest of the images were designated into training and validation sets. Data augmentation of the training and validation sets was performed to increase the number of data sets, reduce model overfitting, and increase the generalization ability of the network model (Fig 3). The images were randomly zoomed (0.8–1.2 times), rotated (-90° to 90°), flipped vertically, flipped horizontally, and cropped (resize after cropping). The number of training and validation sets was increased to train the network models and improve the robustness in practical applications. After data augmentation of the non-test set, the PIQE score was used for evaluation. The images of the lowest 10% scores were chosen as the validation set and the rest as the training set. In the preliminary stage of the study, 5-fold cross-validation had

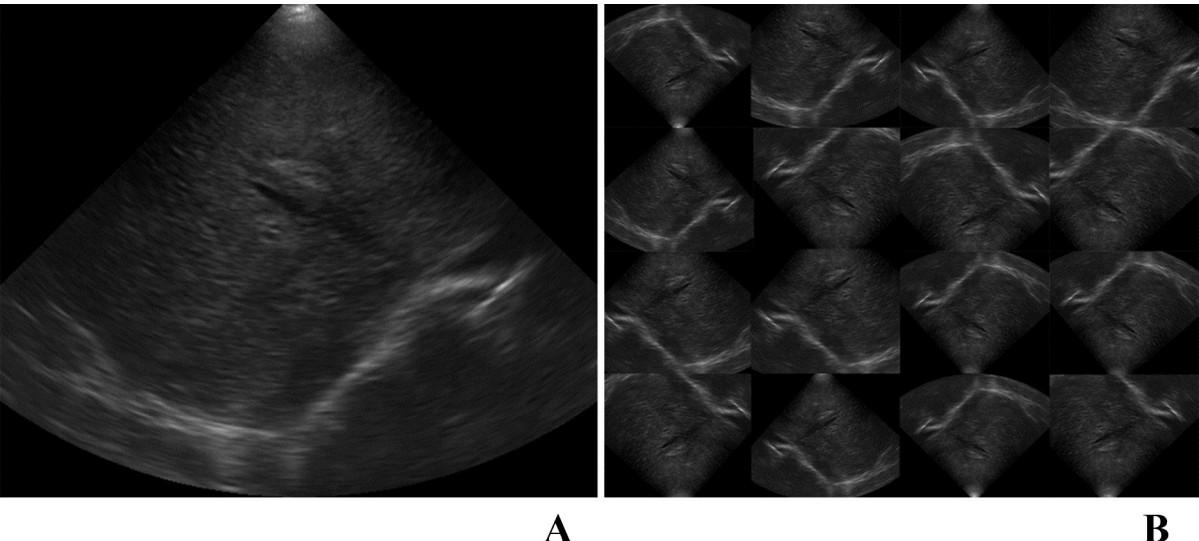

**A** **B**

**Fig 3. Examples of US images from a patient with BA.** A. An original US image. B. The images after data augmentation.

been tried, however, the performance of the models was poor, thus 10-fold cross-validation was used for the final version of the experiment.

## Network model training and classification

As shown in Table 1, a total of 180 patients (41 and 139 patients with and without BA) were included, thus providing a total of 1,976 original US images (417 images of BA and 1,559 images of non-BA). After the adoption of brightness mode and clear images, a total of 1,766 images (350 of BA and 1,416 of non-BA) were included for further study. Among them, 11 patients with BA (143 US images) and 45 patients with non-BA (525 images), a total of 668 original images were initially selected as the test set and excluded as training or validation. After the data augmentation, a total of 21,010 images (3,786 and 17,224 images of BA and non-BA, respectively) were used for training and validation of the study.

In addition, a trilinear interpolation was used to resize the image size to 224 × 224 × 3 before the images were inputted into the network model. The output class was divided into two categories; "with biliary atresia" and "without biliary atresia." Each network model was initialized with Xavier. Moreover, the optimizer used is the SDGM. The initial learning rate was 0.001, and the learning rate was reduced by 0.1 every 10 epochs. The mini-batch size was 25 and a total of 100 epochs were run. All network models were trained using the same training parameters.

## Results

### Performance of CNN models

Under the unified training parameters, data set, and execution environment, the accuracy, F1 score, specificity, false-positive rate (FPR), and false-negative rate (FNR), area under the receiver operating characteristic curve (AUC), and the execution time of each network model were listed in Table 2. The best performance among models was highlighted by bold text.

The channel shuffle processing method of ShuffleNet had good results for the classification of US images for BA diagnosis. The network has up to 90.57% accuracy, 67.83% sensitivity, 32.17% FNR, and the highest F1 score. Furthermore, ResNet-50 and DenseNet-201 presented the best specificity and FPR among the compared models. However, the difference from ShuffleNet was only 0.19% and 0.19% for specificity and FPR, respectively. The execution time of MobileNetV2 is the shortest among all networks, thereby maintaining a certain degree of

**Table 2. Evaluation of the execution results of different network models for the classification of US images.**

| Network | Accuracy | Precision | Sensitivity | Specificity | FPR | FNR | F1 score | AUC | Runtime (s) |
|---------|----------|-----------|-------------|-------------|-----|-----|----------|-----|-------------|
| ResNet-101 | 86.8263% | 78.9474% | 52.4476% | 96.1905% | 3.8095% | 47.5524% | 63.0252% | 84.40% | 11.500042 |
| ResNet-50 | 86.5269% | 81.1765% | 48.2517% | **96.9524%** | **3.0476%** | 51.7483% | 60.5263% | 86.12% | 5.524834 |
| ResNet-18 | 86.8263% | 78.3505% | 53.1469% | 96.0000% | 4.0000% | 46.8531% | 63.3333% | 84.66% | 4.224259 |
| VGG-16 | 85.4790% | 75.5556% | 47.5524% | 95.8095% | 4.1905% | 52.4476% | 58.3691% | 84.55% | 11.426453 |
| VGG-19 | 84.5808% | 73.8095% | 43.3566% | 95.8095% | 4.1905% | 56.6434% | 54.6256% | 83.99% | 19.198317 |
| ShuffleNet | **90.5689%** | **85.0877%** | **67.8322%** | 96.7619% | 3.2381% | **32.1678%** | **75.4864%** | **92.62%** | 4.495658 |
| GoogleNet | 85.3293% | 75.8621% | 46.1538% | 96.0000% | 4.0000% | 53.8462% | 57.3913% | 84.59% | 3.902386 |
| MoblieNetV2 | 85.7784% | 78.5714% | 46.1538% | 96.5714% | 3.4286% | 53.8462% | 58.1498% | 82.21% | **3.852861** |
| DenseNet-201 | 89.2216% | 84.4660% | 60.8392% | **96.9524%** | **3.0476%** | 39.1608% | 70.7317% | 91.85% | 17.020106 |

The results were disclosed by accuracy, precision, sensitivity, and specificity. F1 score was defined as the harmonic mean of precision and recall. FPR: false-positive rate. FNR: false-negative rate. AUC: area under the receiver operating characteristic curve.

accuracy. In addition, the training errors of ResNet-101, ResNet-50, ResNet-18, and VGG-16 were not found during the training process, which caused the overfitting of models. Consequently, the accuracy of the test set decreased. Therefore, the implementation results of each network model will be described in S1 File to understand each network model's ability to identify BA by US images. In general, as the network gets deeper based on the same initial structure, the accuracy should increase accordingly [14]. However, based on the results of ResNet (S1A–S1C Fig), the best performance was provided by ResNet-50 rather than ResNet-101.

All classification network models were used to detect whether the single US image represented BA. For multiple images of one patient, the most intuitive way was to divide the image ($I_{BA}$) identified as biliary atresia by all the input images ($I_{all}$). The Formula was:

$$BA\ Probability = \frac{I_{BA}}{I_{All}}$$

The detailed execution records of ShuffleNet on the test set were the best among the models. Among the 56 patients tested, 44 patients had all US images correctly classified for BA diagnosis (BA for 5 patients and non-BA for 39 patients). The disagreement in individual US image classification was noted in 12 patients, and the patient-by-patient results were listed in Table 3. Images predicted "have" indicated AI diagnosis of BA, and images predicted "none" represented AI excluding BA. The diagnosis was confirmed by IOC for patients with BA and clinical follow-up for patients without BA.

The extremely low accuracy (11.11% and 13.79%) was observed in two patients with BA. The deficiency of AI classification might be due to the few numbers of BA images in the database. Herein, the classification results of US images of "Have02" and "Have06" patients were illustrated. Only one of nine US images of the "Have02" patient was correctly classified by the ShuffleNet model (Fig 4). The only image labeled as "have" was an oblique subcostal scan of the right liver without demonstrating hilum structure but intrahepatic portal vein. The image of the hepatic hilum with a fuzzy triangular cord sign was marked as "none" with a probability of 58.2%. The result indicated that the identification of BA by ShuffleNet could be significantly altered by the clear demonstration of hilum structures.

**Table 3. The identification results of test patients with disagreement of individual US image by ShuffleNet.**

| Patient No. | Diagnosis | Number of images | Images predicted "Have" | Images predicted "None" | Accuracy | BA probability |
|---|---|---|---|---|---|---|
| Have01 | BA | 7 | 4 | 3 | 57.1429% | 57.1429% |
| Have02 | BA | 7 | 6 | 1 | 85.7143% | 85.7143% |
| Have03 | BA | 9 | 1 | 8 | 11.1111% | 11.1111% |
| Have04 | BA | 14 | 13 | 1 | 92.8571% | 92.8571% |
| Have05 | BA | 25 | 23 | 2 | 92.0000% | 92.0000% |
| Have06 | BA | 29 | 4 | 25 | 13.7931% | 13.7931% |
| None01 | Non-BA | 14 | 1 | 13 | 92.8571% | 7.1429% |
| None02 | Non-BA | 8 | 1 | 7 | 87.5000% | 12.5000% |
| None03 | Non-BA | 12 | 7 | 5 | 41.6667% | 58.3333% |
| None04 | Non-BA | 17 | 2 | 15 | 88.2353% | 11.7647% |
| None05 | Non-BA | 15 | 4 | 11 | 73.3333% | 26.6667% |
| None06 | Non-BA | 9 | 2 | 7 | 77.7778% | 22.2222% |

Images predicted "have" indicated AI diagnosis of BA, and images predicted "none" represented AI excluding BA.

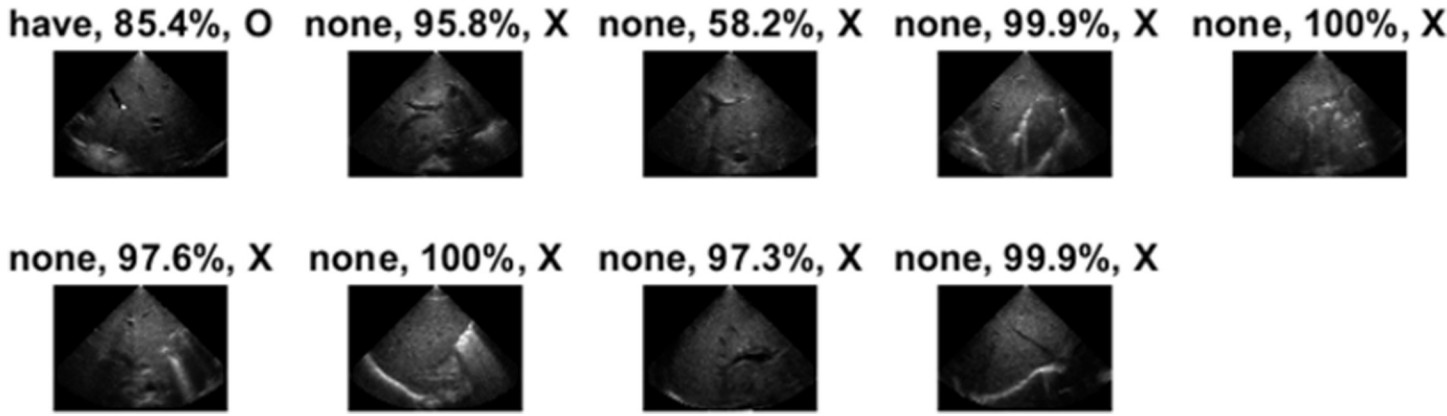

**Fig 4. The detailed identification results of the "Have02" patient with BA.** The first box above the image represents the prediction by the network model. The "have" and the "none" means BA was predicted and the non-BA, respectively. The second box represents the probability of the prediction. Meanwhile, the third box represents whether or not the judgment was correct (O is correct and X is wrong).

Four US images of the "Have06" patient were correctly classified as BA, and the scanned anatomical structures were right side liver parenchyma, gallbladder, and hilum structure (Fig 5). Like the "Have02" patient, images of the right liver revealed diagnostic value in US-based AI model classification. A total of 25 images of the "Have06" patient were identified as the non-BA with a high probability (65.5%–100%). The results showed the incapability to achieve correct prediction based on a single network model, like ShuffleNet for the "Have06" patient.

## Methods to resolve disagreement by individual US images

Two feasible methods were proposed based on the patient-based experimental results to decrease the effect of the outliers. The first method was the positive-dominance law. That was, for a single patient, if the CNN model predicted any input B-mode US as BA, the patient would be classified as "BA." The prediction results of the test set by ShuffleNet following positive-dominance law were listed in Table 4. However, this method might lead to an increase in FPR of 13.64% in this case.

For a single patient, the weighted average of the probability of BA recognized by the network was calculated based on the prediction results of the input B-mode US images. The patient was then classified as BA if the value was greater than a certain threshold (20% in this case). The method was named thresholding law and the results of ShuffleNet were listed in Table 4. In the preliminary stage of the study, thresholds of 10%, 20%, and 30% were used, and the final threshold was set as 20% for better performance. Herein, the higher threshold would further compromise the sensitivity of the AI tool. The results of thresholding law were improved by accuracy, precision, specificity, and FPR as compared with positive-dominance law.

## Application for upcoming patients

There were three patients suspected of having BA by two authors (S.Y.H. and C.M.C.) within one month after the network training. Some trained models tested raw images of hepatobiliary US. The prediction results and final diagnoses were listed in Table 5. The final diagnoses were confirmed by IOC. For Case 1, only the ShuffleNet strongly predicted BA (93.75%), and most models made the correct prediction as to the final diagnosis. Meanwhile, for Case 2, only the ShuffleNet predicted BA, and for Case 3, ShuffleNet and GoogleNet predicted BA. The latter

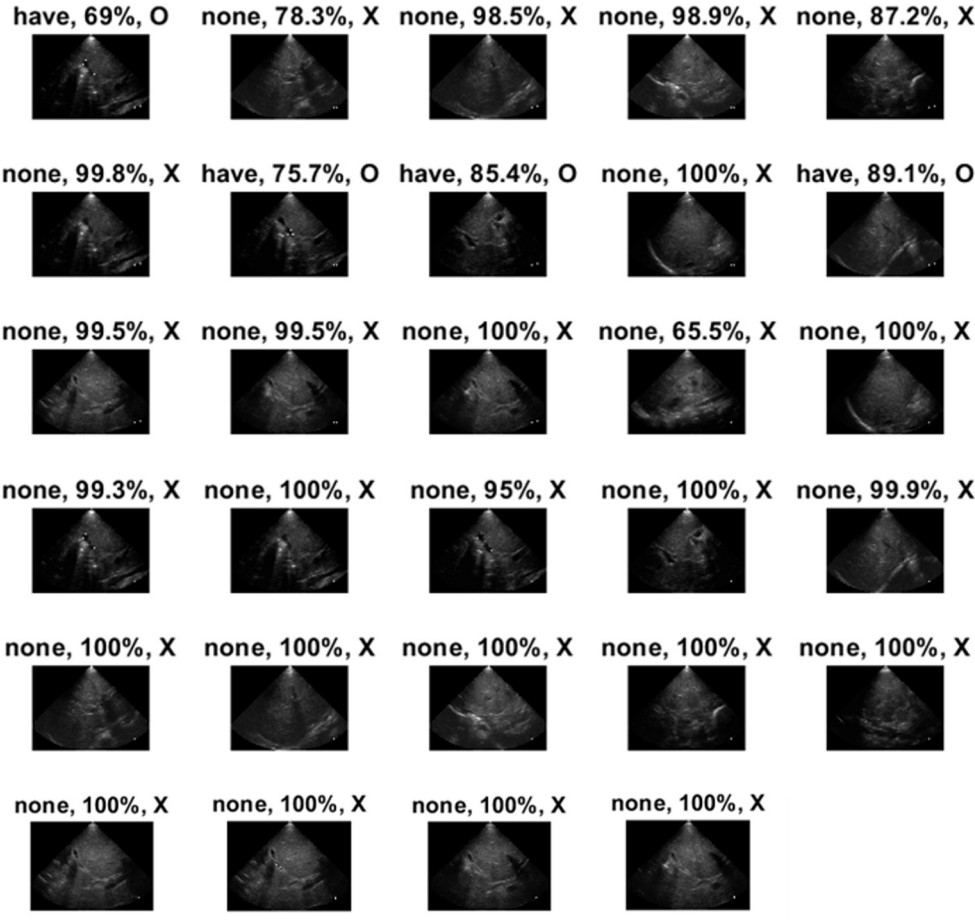

**Fig 5. The detailed identification results of the "Have06" patient with BA.** The first box above the image represents the prediction by the network model. The "have" and the "none" means BA was predicted and the non-BA, respectively. The second box represents the probability of the prediction. Meanwhile, the third box represents whether or not the judgment was correct (O is correct and X is wrong).

two patients were confirmed to have BA by IOC and then underwent simultaneous laparoscopic Kasai portoenterostomy.

## Discussion

Image classification is a typical task in computer vision, and the output is the category and probability of the images. Object recognition and detection identify the category and location in the images. A bounding box is often used to output the detected category, probability, and location information. A convolutional neural network (CNN) is the most popular solution for image classification. In addition, various CNN models are applied to medical US images, whether a single model or a combination of different network models for classification and prediction, respectively [15, 16].

In the field of image classification, the residual network (ResNet) [17, 18] is often the first choice for its simple and practically deeper construction. Moreover, as compared with AlexNet [19], VGGNet [20], and GoogLeNet [21], ResNet has proven that as the depth of the network increases, the accuracy and other indicators of the network should be increased instead of deteriorating or gradient vanishing. However, the theoretical benefits provided by the deeper

**Table 4. Confusion matrix and execution results of test set by ShuffleNet obeying different laws of interpretation.**

| Positive-dominance law | | | | | Thresholding law | | | | |
| --- | --- | --- | --- | --- | --- | --- | --- | --- | --- |
| Confusion Matrix by Patients | | | Accuracy | 89.09% | Confusion Matrix by Patients | | | Accuracy | 90.91% |
| Actual / Predicted | BA | Non-BA | Precision | 64.71% | Actual / Predicted | BA | Non-BA | Precision | 75.00% |
| | | | Sensitivity | 100.00% | | | | Sensitivity | 81.82% |
| BA Patients | 11 | 0 | Specificity | 86.36% | BA Patients | 9 | 2 | Specificity | 93.18% |
| | | | FPR | 13.64% | | | | FPR | 6.82% |
| Non-BA Patients | 6 | 38 | FNR | 0.00% | Non-BA Patients | 3 | 41 | FNR | 18.18% |
| | | | F1-Score | 78.57% | | | | F1-Score | 78.26% |

Positive-dominance law indicated that the classification would be BA even if only one US image was predicted positive. Thresholding law indicated BA if more than 20% of images were predicted positive.

construction of ResNet were not shown in this study, and overfitting is assumed to be the reason. Calculations of MobileNetV1 would be less than half of the original convolution structure [22–24]. MobileNetV2 is based on MobileNetV1, and linear bottlenecks and inverted residuals are added [25] to reduce the information loss caused by rectified linear units (ReLU). In this study, the shortest runtime was created by MobileNetV2. SqueezeNet is a lightweight network model, and the size of parameters is only 2.14% of AlexNet with the equivalent performance [26]. ShuffleNet is based on SqueezeNet with some changes [27]. The ShuffleNetV1 architecture refers to the ResNet bottleneck design. Furthermore, the group convolution and channel shuffle are used to compress calculations, exchange information between channels, and learn more complex features. ShuffleNetV2 avoids the increasing multiply-accumulate (MAC) of many 1 × 1 pointwise convolutions as compared with ShuffleNetV1 [28]. Therefore, choosing an appropriate lightweight CNN is vital for further study. A simple comparison between MobileNetV2 and ShuffleNetV2 according to the work of Ma et al. in 2018 [28] proved that ShuffleNetV2 provides less amount calculation and requires few parameters. However, the Top-1 accuracy of all listed models by Ma *et al.* is not satisfying. The Top-5 accuracy of ShuffleNetV2 decreases from 92.4% to 83.6%, following the trend of structural complexity.

Meanwhile, transfer learning effectively reduces network model overfitting, solves time-consuming problems, and prevents cumbersome data labeling and complex data acquisition [29, 30]. The practical application of image classification could be more effective, robust, and extensive. However, accuracy in medical imaging is the most critical outcome, particularly for image recognition and classification of diagnosis. Based on the known poor accuracy, the pre-

**Table 5. Prediction results for BA of three successive patients.**

| | Case 1 | Case 2 | Case 3 |
| --- | --- | --- | --- |
| Network | BA Probability | BA Probability | BA Probability |
| ResNet-101 | 56.25% | 23.53% | 46.15% |
| VGG-16 | 12.50% | 29.41% | 23.08% |
| ShuffleNet | 93.75% | 88.24% | 100.00% |
| GoogleNet | 18.75% | 52.94% | 69.23% |
| MobileNetV2 | 31.25% | 47.06% | 46.15% |
| DenseNet-201 | 31.25% | 29.41% | 53.85% |
| Final Diagnosis | Non-BA | BA | BA |

All three cases were clinically suspected to have a BA and the final diagnosis was achieved by IOC.

trained network by ImageNet would be abandoned in the preliminary stage of this study. Instead, training new models would be initiated from scratch. The trained network models would be tested, compared, and modified in the following stage.

Among patients with prolonged jaundice, BA could be suspected by the shape of the hepatic portal (vascular and bile duct entrance) and gallbladder morphology in US images. The contribution of deep learning to the classification was significant. Thus far, deep learning has achieved rapid development in network architecture or models, such as deeper network architectures and deep generative models. There have been various papers on applying deep learning in medical image analysis. These reports focused on the entire field of medical image analysis, like US, MRI, or computerized tomography (CT) scan [15, 16, 29, 30].

Zhou et al. [11] used CNN models to detect BA from US gallbladder and hilum bile duct structure images. The patient-level sensitivity and specificity were 93.1% and 93.9%, respectively. Herein, an AUC of 0.956 was achieved. However, the results were contributed by expertise (human or machine) cropping images, and the images of liver parenchyma and surrounding anatomical structures were not considered. For a regular abdominal US examination, as many as 10–20 images would be acquired, particularly when the clinical suspicion was unclear. In addition, by focusing on the critical structures of BA, like gallbladder and bile ducts, overfitting of the model might be an issue. Furthermore, the visualization of the gallbladder in US images was required in the data set; thus, patients without fasting US images might be lost. In the outpatient clinic setting, screening US study would often be performed randomly even after feeding. In this study, no limitation of feeding conditions was set for better simulation of the clinical scenario before the US image is obtained. Kuo et al. [10] developed an AI method for estimating glomerular filtration rate by renal US images without preprocessing image identification or labeling. The concept was adopted in the study, and deep learning models were to find the classification of BA or non-BA based on less selected and random hepatobiliary US images. Meanwhile, two methods were proposed to solve the disagreement of image-based prediction. For thresholding law, the threshold value was set by researchers without supporting evidence, and the value might be changed by increasing the size of the database. Though the thresholding law decreased FPR by 6.82%, the high FNR of 18.18% was not acceptable in clinical application. The prognosis of BA has known to be significantly influenced by the time of intervention [1, 2]. Therefore, the early diagnosis was crucial for BA patients. The main requirement of the screening tool should be high sensitivity. The FPR of 13.64% by positive-dominance law meant that one or two of every 10 patients who underwent IOC would eventually be confirmed with a non-BA diagnosis. The negative results of IOC to exclude BA are acceptable for pediatricians and pediatric surgeons. In addition, the clinicians could make more precise diagnoses by incorporating US-based AI classification with stool color, laboratory data, and other image studies.

Pathological jaundice of infants caused by BA is a critical clinical condition. For now, invasive examination methods such as liver biopsy and IOC are still the golden standard for diagnosis. This study explored the possibility of non-invasive diagnostic methods using hepatobiliary US images based on deep learning methods. The experimental results showed that the construction of the ShuffleNet network was excellent at processing US images for diagnosis of BA, with an accuracy of 90.57%. Thus, the results of real-world data revealed that only the ShuffleNet could predict BA with a higher probability. Moreover, ShuffleNet had the advantages of rapid processing and a lightweight model. Further application in community clinics and rural regions would be more feasible. All hepatobiliary US images were included in the database of this study, and the generalization might prevent expertise selection before image processing and overfitting of model training. Based on US images of disagreement prediction, right liver parenchyma morphology seemed important for BA diagnosis in addition to the gallbladder and hepatic hilum.

The main limitations of this study were the size of the US image database, the doubt of overfitting, and more requirements of test sets for verification. In addition, the overdiagnosis of BA by ShuffleNet was an issue. The lower sensitivity of the current models had left room for further perfection by adding more images of the training set. However, the reported high sensitivity was achieved by experts of pediatric hepatobiliary US image reading. In the real world, the diagnosis is often delayed by inexperienced physicians, and this study verified the method's feasibility to assist the screening of BA by US-based deep learning models. Finally, the current study only calculated the different diagnostic results of ShuffleNet following positive-dominance and thresholding laws, the two methods dealing with classification disagreement, and the calculation of other models should be further performed.

## Conclusion

ShuffleNet has the best prediction performance among tested CNN models for diagnosing BA from ultrasound images. The disagreement of individual US images of one patient could be resolved by applying the positive-dominance law. US images of the right liver are potential roles in screening the BA. Moreover, the lightweight model has a promising application for non-experts in local clinics or rural regions.

## Supporting information

**S1 Fig. Performance of CNN models.** A, Confusion matrix and ROC curve of ResNet-101. B, Confusion matrix and ROC curve ResNet-50. C, Confusion matrix and ROC curve ResNet-18. D, Confusion matrix and ROC curve of VGG-16. E, Confusion matrix and ROC curve of VGG-19. F, Confusion matrix and ROC curve of ShuffleNet. G, Confusion matrix and ROC curve of GoogleNet. H, Confusion matrix and ROC curve of MobileNetV2. I, Confusion matrix and ROC curve of DenseNet-201. AUC: area under curve.
(TIF)

**S1 Table. Comparison of AUC between models by Delong test.** The *p* values in bold texts indicate significant difference.
(DOCX)

**S1 File. Implementation results of each network model used in this study.** Details of execution results are described.
(DOCX)

**S1 Graphical abstract.**
(TIF)

## Author Contributions

**Conceptualization:** Fang-Rong Hsu, Sheng-Yang Huang.

**Data curation:** Sheng-Tong Dai, Chia-Man Chou, Sheng-Yang Huang.

**Formal analysis:** Fang-Rong Hsu, Sheng-Tong Dai.

**Funding acquisition:** Fang-Rong Hsu, Chia-Man Chou.

**Investigation:** Sheng-Tong Dai, Sheng-Yang Huang.

**Methodology:** Fang-Rong Hsu, Chia-Man Chou, Sheng-Yang Huang.

**Project administration:** Fang-Rong Hsu, Chia-Man Chou, Sheng-Yang Huang.

**Resources:** Fang-Rong Hsu, Sheng-Yang Huang.

**Software:** Fang-Rong Hsu, Sheng-Tong Dai.

**Supervision:** Fang-Rong Hsu, Chia-Man Chou.

**Validation:** Chia-Man Chou, Sheng-Yang Huang.

**Visualization:** Sheng-Yang Huang.

**Writing – original draft:** Fang-Rong Hsu, Sheng-Tong Dai.

**Writing – review & editing:** Sheng-Yang Huang.

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
