## [Decision Letter · Decision Letter 0]

12 Jul 2022

PONE-D-22-08504The application of artificial intelligence to support biliary atresia diagnosis by ultrasound images: A study based on deep learning modelsPLOS ONE

Dear Dr. Huang,

Thank you for submitting your manuscript to PLOS ONE. After careful consideration, we feel that it has merit but does not fully meet PLOS ONE’s publication criteria as it currently stands. Therefore, we invite you to submit a revised version of the manuscript that addresses the points raised during the review process.

We look forward to receiving your revised manuscript.

Kind regards,

Gregory Tiao, M.D.

Academic Editor

PLOS ONE

Journal Requirements:

Reviewers' comments:

Reviewer's Responses to Questions

**Comments to the Author**

1. Is the manuscript technically sound, and do the data support the conclusions?

Reviewer #1: Partly

2. Has the statistical analysis been performed appropriately and rigorously? 

Reviewer #1: I Don't Know

3. Have the authors made all data underlying the findings in their manuscript fully available?

Reviewer #1: Yes

4. Is the manuscript presented in an intelligible fashion and written in standard English?

Reviewer #1: No

5. Review Comments to the Author

Reviewer #1: This study aimed to establish an ultrasound-based deep learning model for biliary atresia diagnosis. Several neural networks were used for model building. The authors concluded that ShuffleNet had the best prediction performance among tested CNN models for diagnosing BA from ultrasound images, and the lightweight model had a promising application for non-experts in local clinics or rural regions. Generally, I think this manuscript is clinically meaningful, but the methodological part is not detailed enough and the small sample size used in the study limits the generalization of their findings.

My concerns are the following.

1. Generally, the whole manuscript need both language editing and scientific editing. For example, the introduction is a little too long. Plus, the Result and Discussion focus too much on different deep learning network models. Since most of the potential readers of the paper are radiologist, pediatric surgeons and general physician, who might be not interested in deep learning technologies, the authors need to cut down some of these contents(Or presented as suppentary materials).

2. Please include a table of patient demographics/characteristics (including age, gender, serum liver function test, etc).

3. Inclusion and exclusion criteria for ultrasound images not specifically mentioned in the methodology. In addition, the authors did not mention whether each image involved in this study contained specific structures, such as gallbladder, portal vein, common bile duct, etc.

4. What kind of Ultrasound scanner with what kind of probe/transducer was used to obtain all the images?

5. Page 15, what is “NPO” for?

6. 4. Page 17, third paragraph: “Some images were randomly selected as the test set among the included US images.” Could you be more specific about the sampling strategy for training set, validation set, test set?

7. Page 17, third paragraph: “After data augmentation of the non-test set, the PIQE score was used for evaluation. The images of the lowest 10% scores were chosen as the validation set and the rest as the training set.” Why the images of lowest 10% scores were chosen as validation set? Is there any reason to divide the training set and the validation set after data augmentation?

8. In Table 2, the AUC value of each model should be added and it is better to perform a Delong test to compare the performances among models.

9. Page 24, second paragraph: “Four US images of the “Have06” patient were classified correctly as BA, and the regions of interest were right side liver parenchyma, gallbladder, and hilum structure (Fig. 6).” How did you define the region of interest for the image? Did you do class activation map(CAM) test?

10. Page 27, line 3-4: “If the value was greater than a certain threshold (20% in this case), the patient was then classified as BA.” How 20% of this study was determined?

11. In clinical practice, the sensitivity of ultrasound in diagnosing biliary atresia can be as high as 80%, but the models used in this study had the highest sensitivity of only 67.8322%. Due to the potential harm caused by delayed diagnosis of biliary atresia, the low-sensitivity model lacks clinical application value.

12. Page 33, line 1-4: “According to US images of disagreement prediction, right liver parenchyma morphology was important for BA diagnosis in addition to the gallbladder and hepatic hilum”. The authors did not provide sufficient evidence to confirm their theory( the right liver parenchyma morphology is helpful in the diagnosis of biliary atresia), and it is inappropriate to draw this conclusion.

13. Page 32, line 3-4: “Two methods to solve disagreement of image-based prediction were proposed.” This study only calculated the diagnostic results of the two methods in the ShuffleNet model, which should be supplemented with the calculation of other models, and perhaps a more ideal effect may be obtained.

6. PLOS authors have the option to publish the peer review history of their article (what does this mean?). If published, this will include your full peer review and any attached files.

Reviewer #1: **Yes: **Luyao Zhou

---

## [Author Response · Author response to Decision Letter 0]

21 Aug 2022

1. Generally, the whole manuscript need both language editing and scientific editing. For example, the introduction is a little too long. Plus, the Result and Discussion focus too much on different deep learning network models. Since most of the potential readers of the paper are radiologist, pediatric surgeons and general physician, who might be not interested in deep learning technologies, the authors need to cut down some of these contents (Or presented as suppentary materials).

Response: 

The manuscript has been through an additional language review after the revisions. The introduction, result, and discussion have been cut down, especially the contents of deep learning network models. The reduced information has been moved to Supplementary Materials.

2. Please include a table of patient demographics/characteristics (including age, gender, serum liver function test, etc).

Response: 

The information has been added in Table 1.

3. Inclusion and exclusion criteria for ultrasound images not specifically mentioned in the methodology. In addition, the authors did not mention whether each image involved in this study contained specific structures, such as gallbladder, portal vein, common bile duct, etc.

Response: 

The criteria have been added in “2.2 Data Collection and Labeling”, lines 160-166.

The description of anatomy is added on lines 185-189.

4. What kind of Ultrasound scanner with what kind of probe/transducer was used to obtain all the images?

Response: 

The information is provided on lines 189-192.

5. Page 15, what is “NPO” for?

Response: 

NPO stands for fasting and has been clarified on lines 196-197.

6. Page 17, third paragraph: “Some images were randomly selected as the test set among the included US images.” Could you be more specific about the sampling strategy for the training set, validation set, test set?

Response: 

A blinded author performs the sampling, and the method is described on lines 229-231.

7. Page 17, third paragraph: “After data augmentation of the non-test set, the PIQE score was used for evaluation. The images of the lowest 10% scores were chosen as the validation set and the rest as the training set.” Why the images of lowest 10% scores were chosen as validation set? Is there any reason to divide the training set and the validation set after data augmentation?

Response: 

In the preliminary stage of the study, 5-fold cross-validation was tried, but the performance of the models was poor. Thus, 10-fold cross-validation was used for the final version of the experiment. The condition has been added on lines 241-243.

8. In Table 2, the AUC value of each model should be added, and it is better to perform a Delong test to compare the performances among models.

Response: 

The AUC of each model has been added in Table 2, and Delong test results are placed in Supplementary Materials.

9. Page 24, second paragraph: “Four US images of the “Have06” patient were classified correctly as BA, and the regions of interest were right side liver parenchyma, gallbladder, and hilum structure (Fig. 6).” How did you define the region of interest for the image? Did you do class activation map(CAM) test?

Response: 

The term ROI is not current in this sentence and has been changed to “scanned anatomical structures” in line 320. ROI definition is described on lines 198-205.

10. Page 27, line 3-4: “If the value was greater than a certain threshold (20% in this case), the patient was then classified as BA.” How 20% of this study was determined?

Response: 

In the preliminary stage of the study, thresholds of 10%, 20%, and 30% were used, and the final threshold was set as 20% for better performance. The description is added on lines 342-344.

11. In clinical practice, the sensitivity of ultrasound in diagnosing biliary atresia can be as high as 80%, but the models used in this study had the highest sensitivity of only 67.8322%. Due to the potential harm caused by delayed diagnosis of biliary atresia, the low-sensitivity model lacks clinical application value.

Response: 

The lower sensitivity by the current models is the limit of this study and has been added in the discussion on lines 458-463. However, the authors believed that the reported high sensitivity was achieved by pediatric hepatobiliary US image reading experts. In the real world, the diagnosis is often delayed by inexperienced physicians. This study verified the method’s feasibility in assisting the screening of BA by US-based deep learning models.

12. Page 33, line 1-4: “According to US images of disagreement prediction, right liver parenchyma morphology was important for BA diagnosis in addition to the gallbladder and hepatic hilum”. The authors did not provide sufficient evidence to confirm their theory (the right liver parenchyma morphology is helpful in the diagnosis of biliary atresia), and it is inappropriate to draw this conclusion.

Response: 

The authors agreed that the conclusion was not objective, and the sentence has been rephrased on lines 453-455.

13. Page 32, line 3-4: “Two methods to solve disagreement of image-based prediction were proposed.” This study only calculated the diagnostic results of the two methods in the ShuffleNet model, which should be supplemented with the calculation of other models, and perhaps a more ideal effect may be obtained.

Response: 

Lacking further applications of two methods of solving disagreement for other networks is the limit of this study and has been added in the discussion on lines 463-466.

---

## [Decision Letter · Decision Letter 1]

4 Oct 2022

The application of artificial intelligence to support biliary atresia screening by ultrasound images: A study based on deep learning models

PONE-D-22-08504R1

Dear Dr. Huang,

We’re pleased to inform you that your manuscript has been judged scientifically suitable for publication and will be formally accepted for publication once it meets all outstanding technical requirements.

Kind regards,

Gregory Tiao, M.D.

Academic Editor

PLOS ONE

Additional Editor Comments (optional):

The authors have addressed the concerns raised by previous reivew

Reviewers' comments:

Reviewer's Responses to Questions

**Comments to the Author**

1. If the authors have adequately addressed your comments raised in a previous round of review and you feel that this manuscript is now acceptable for publication, you may indicate that here to bypass the “Comments to the Author” section, enter your conflict of interest statement in the “Confidential to Editor” section, and submit your "Accept" recommendation.

Reviewer #1: All comments have been addressed

2. Is the manuscript technically sound, and do the data support the conclusions?

Reviewer #1: Yes

3. Has the statistical analysis been performed appropriately and rigorously? 

Reviewer #1: Yes

4. Have the authors made all data underlying the findings in their manuscript fully available?

Reviewer #1: Yes

5. Is the manuscript presented in an intelligible fashion and written in standard English?

Reviewer #1: Yes

6. Review Comments to the Author

Reviewer #1: Since all my comments have been well-responded, I recommend to accept this manuscript for publishing.

7. PLOS authors have the option to publish the peer review history of their article (what does this mean?). If published, this will include your full peer review and any attached files.

Reviewer #1: No

---

## [Editor Report · Acceptance letter]

10 Oct 2022

PONE-D-22-08504R1 

The application of artificial intelligence to support biliary atresia screening by ultrasound images: A study based on deep learning models 

Dear Dr. Huang:

I'm pleased to inform you that your manuscript has been deemed suitable for publication in PLOS ONE. Congratulations! Your manuscript is now with our production department. 

Kind regards, 

on behalf of

Dr. Gregory Tiao 

Academic Editor

PLOS ONE